# Outcomes of Kidney Transplantation in Fabry Disease: A Meta-Analysis

**DOI:** 10.3390/diseases9010002

**Published:** 2020-12-23

**Authors:** Maria L. Gonzalez Suarez, Charat Thongprayoon, Panupong Hansrivijit, Juan Medaura, Pradeep Vaitla, Michael A. Mao, Tarun Bathini, Boonphiphop Boonpheng, Swetha R. Kanduri, Karthik Kovvuru, Arpita Basu, Wisit Cheungpasitporn

**Affiliations:** 1Division of Nephrology and Hypertension, Department of Medicine, Mayo Clinic, Rochester, MN 55905, USA; malourdes.gonzalez@gmail.com; 2Division of Nephrology, Department of Internal Medicine, University of Mississippi Medical Center, Jackson, MS 39216, USA; jmedaura@umc.edu (J.M.); pvaitla@umc.edu (P.V.); 3Department of Internal Medicine, UPMC Pinnacle, Harrisburg, PA 17105, USA; hansrivijitp@upmc.edu; 4Division of Nephrology and Hypertension, Department of Medicine, Mayo Clinic, Jacksonville, FL 32224, USA; mao.michael@mayo.edu; 5Department of Internal Medicine, University of Arizona, Tucson, AZ 85721, USA; tarunjacobb@gmail.com; 6Department of Medicine, David Geffen School of Medicine, University of California, Los Angeles, CA 90095, USA; boonpipop.b@gmail.com; 7Division of Nephrology, Ochsner Medical Center, New Orleans, LA 70121, USA; svetarani@gmail.com (S.R.K.); karthik.kovvuru@ochsner.org (K.K.); 8Emory Transplant Center and Department of Medicine, Renal Division, Emory University School of Medicine, Atlanta, GA 30322, USA; arpita.basu@emory.edu

**Keywords:** Fabry disease, kidney transplant, kidney transplantation, meta-analysis, systematic review

## Abstract

Background: Fabry disease (FD) is a rare X-linked lysosomal storage disorder with progressive systemic deposition of globotriaosylceramide, leading to life-threatening cardiac, central nervous system, and kidney disease. Current therapy involves symptomatic medical management, enzyme replacement therapy (ERT), dialysis, kidney transplantation, and, more recently, gene therapy. The aim of this systematic review was to assess outcomes of kidney transplantation among patients with FD. Methods: A comprehensive literature review was conducted utilizing MEDLINE, EMBASE, and Cochrane Database, from inception through to 28 February 2020, to identify studies that evaluate outcomes of kidney transplantation including patient and allograft survival among kidney transplant patients with FD. Effect estimates from each study were extracted and combined using the random-effects generic inverse variance method of DerSimonian and Laird. Results: In total, 11 studies, including 424 kidney transplant recipients with FD, were enrolled. The post-transplant median follow-up time ranged from 3 to 11.5 years. Overall, the pooled estimated rates of all-cause graft failure, graft failure before death, and allograft rejection were 32.5% (95%CI: 23.9%–42.5%), 14.5% (95%CI: 8.4%–23.7%), and 20.2% (95%CI: 15.4%–25.9%), respectively. In the sensitivity analysis, limited only to the recent studies (year 2001 or newer when ERT became available), the pooled estimated rates of all-cause graft failure, graft failure before death, and allograft rejection were 28.1% (95%CI: 20.5%–37.3%), 11.7% (95%CI: 8.4%–16.0%), and 20.2% (95%CI: 15.5%–26.0%), respectively. The pooled estimated rate of biopsy proven FD recurrence was 11.1% (95%CI: 3.6%–29.4%), respectively. There are no significant differences in the risks of all-cause graft failure (*p* = 0.10) or mortality (0.48) among recipients with vs. without FD. Conclusions: Despite possible FD recurrence after transplantation of 11.1%, allograft and patient survival are comparable among kidney transplant recipients with vs. without FD.

## 1. Introduction

Fabry disease (FD) is a rare, progressive, multisystemic, and X-linked inherited lysosomal disorder, caused by genetic variations in GLA (HUGO Gene Nomenclature Committee ID: 4296; Gene Entrez: 2717; NCBI reference sequence: NM_000169.2), which encodes α-galactosidase (α-Gal, Enzyme Commission number: EC 3.2.1.22; UniProt ID: P06280) [1,2]. Studies have recently identified a variety of variants underlying the phenotypic heterogeneity of this genetic disorder [3,4,5,6,7,8,9,10,11,12,13,14,15,16,17,18,19]. The GLA variants can lead to the deficiency of the lysosomal enzyme ɑ-galactosidase A (α-gal A) [20,21,22,23,24,25,26,27]. This enzyme is active in glycosphingolipid catabolism and accumulation of neutral glycosphingolipids, namely, globotriaosylceramide (Gb3), which is the hallmark pathogenesis leading to clinical syndromes [20,28]. The incidence of FD in the general population ranges from ~1:40,000 to 1:60,000 among male patients [29,30,31]. The clinical presentations of Fabry syndrome include cardiomyopathy, cerebrovascular disease, and renal failure [32,33]. Renal involvement in FD has been recognized as a cardinal feature which pertains specific pathological characteristics and prognosis [34].

Fabry nephropathy is one of the causes of chronic kidney disease, similar to diabetic kidney disease, which could progress to end-stage kidney disease in the fifth decade of life [34]. It has been reported that the natural history of Fabry nephropathy may evolve in three clinical phases [35]. The first phase is glomerular hyperfiltration which usually has onset in childhood or adolescence. The second phase involves proteinuria, lipiduria, or Maltese cross crystals. In Branton et al., proteinuria is the most common renal manifestation at the mean age of 34 years [34]. Lastly, the final phase is characterized by several renal diseases and progression to end-stage kidney disease. Patients may also exhibit vascular, cardiac and cerebrovascular involvement at this stage as well. Other uncommon features of FD include, but are not limited to, nephrogenic diabetes insipidus and Fanconi syndrome. Current therapies involve symptomatic medical management, enzyme replacement therapy (ERT), dialysis, and kidney transplantation [35].

The role of kidney transplantation in FD has been subjected to certain controversies. Some evidence suggests that kidney transplant increases serum α-gal A [36,37,38] and urine α-gal A [39,40]. Graft failure secondary to substrate deposition has also been suggested [41]. However, current evidence to date on the outcomes of kidney transplantation in FD is limited due to the small sample size in each individual study making it difficult to apply to clinical practice. Thus, this systematic review and meta-analysis was conducted to describe the clinical outcomes among the pooled sample size from all available studies.

## 2. Materials and Methods

### 2.1. Search Strategies

A comprehensive search of several databases, from each database’s inception to February 28, 2020, was conducted. The databases included OVID MEDLINE (1946 to February 2020), EMBASE (1988 to February 2020), and the Cochrane Database of Systematic Reviews (database inception to February 2020). The systematic literature review was conducted independently by two investigators (M.L.G.S. and C.T.) using the search strategy that consolidated the terms of (“kidney transplantation” OR “kidney graft” OR “kidney graft rejection” OR (renal AND transplantation)) AND (“fabry disease” OR “lysosomal storage disease”). The actual strategy listing all search terms used is available in the online Appendix A. There were no restrictions on language, sample size, or study duration. This study was conducted by the Preferred Reporting Items for Systematic Reviews and Meta-Analysis (PRISMA) statement [42].

### 2.2. Study Selection

Eligible studies must be clinical trials, observational studies (cohort, case-control, or cross-sectional studies) that reported incidence and/or outcomes of kidney transplantation among patients with FD. Retrieved articles were individually reviewed for eligibility by two investigators (M.L.G.S. and C.T.). Discrepancies were addressed and resolved by a third investigator (W.C.). Inclusion was not limited by language, age, sample size, or study duration. 

### 2.3. Data Extraction

The following data were extracted: first author name, year of publication, number of patients, duration of follow-up, diagnosis of FD, mean age, sex, incidence of FD recurrence, patient survival, graft survival, and allograft rejection. The primary outcome was allograft survival.

### 2.4. Data Synthesis and Statistical Analysis

We calculated pooled estimated rates of FD recurrence, patient survival, graft survival, and allograft rejection among kidney transplant patients with FD. A random-effects model was used due to the expected clinical heterogeneity in the included populations [43]. All pooled estimates were shown with 95% confidence intervals (CIs). Heterogeneity among effect sizes estimated by individual studies was described with the I^2^ statistic and the chi-square test. A value of I2 of 0% to 25% represents insignificant heterogeneity, 26% to 50% low heterogeneity, 51% to 75% moderate heterogeneity, and 76 to 100% high heterogeneity [44].

Publication bias was evaluated using the Egger test [45]. A *p*-value of less than 0.05 indicates the presence of publication bias. The meta-analysis was performed by the Comprehensive Meta-Analysis 3.3 software (Biostat Inc., Englewood, NJ, USA). The data for this meta-analysis are publicly available through the Open Science Framework (URL: osf.io/amrxq/).

## 3. Results

In total, 11 studies, including 424 kidney transplant recipients with FD, were enrolled (Figure 1 and Table 1) [46,47,48,49,50,51,52,53,54,55,56]. The post-transplant median follow-up time ranged from 3 to 11.5 years.

Overall, the pooled estimated rates of all-cause graft failure, graft failure before death, and allograft rejection were 32.5% (95%CI: 23.9%–42.5%, I^2^ = 50%), 14.5% (95%CI: 8.4%–23.7%), and 20.2% (95%CI: 15.4%–25.9%), respectively (Figure 2). From a sensitivity analysis limited only to the recent studies (year 2001 or newer when ERT became available) (Table 1), the pooled estimated rates of all-cause graft failure, graft failure before death, and allograft rejection were 28.1% (95%CI: 20.5%–37.3%), 11.7% (95%CI: 8.4%–16.0%), and 20.2% (95%CI: 15.5%–26.0%), respectively.

The pooled estimated rate of biopsy proven FD recurrence was 11.1% (95%CI: 3.6%–29.4%), respectively. There are no significant differences in the risks of all-cause graft failure (P = 0.10) (Figure 3) or mortality (0.48) among recipients with vs. without FD. The Egger’s regression demonstrated no significant publication bias for all analyses (*p* > 0.05).

## 4. Discussion

Here, we reported that the pooled estimated rates of all-cause graft failure, graft failure before death, and allograft rejection were 32.5%, 14.5%, and 20.2%, respectively, with a median follow-up time ranging from 3 to 11.5 years. We also found no significant difference in the risk of all-cause graft failure or the mortality among recipients with and without FD. However, the incidence of graft failure in FD patients is overtly higher than those with diabetes. One retrospective cohort study showed that the incidence of graft failure in diabetic patients was 6.7% compared to 2.8% in nondiabetic patients [57]. Overall, our data indicated that kidney transplantation due to FD is safe and should be considered in patients with end-stage kidney disease due to Fabry nephropathy.

In 1975, the American College of Surgeons and the National Institute of Health Renal Transplant Registry reviewed the outcomes of kidney transplantation in several congenital metabolic diseases. The good outcomes following kidney transplantation were highlighted in patients with Alport syndrome, amyloidosis, cystinosis, and others with the exception of FD and oxalosis. The graft survival after one year was only 33% without reported evidence of relapses [46]. The 5-year survival rate among kidney transplant recipients with FD was up to 26% with a high incidence of sepsis [58]. Because of these discouraging early data, kidney transplants in FD patients were not recommended.

We also demonstrated that the incidence of all-cause graft failure (28.1%) and death-censored graft failure (11.7%) have declined over time, especially in studies published after 2001 or when enzyme replacement therapy (ERT) became available, whereas the incidence of allograft rejection remained unchanged (20.2%). Earlier studies among Fabry patients after kidney transplantation reported a rise in galactosidase enzyme levels secondary to lysosomal enzyme release from transplanted organ [59,60,61]. However, it was reported to be from increased α-N-acetylgalactosaminidase (α-Gal B) activity, a similar enzyme to α-Gal A [60,61]. Subsequently, the effect was found to be transient and did not effectively reduce Gb3 levels. Hence, further ERT is warranted after kidney transplantation [62,63]. Cybulla et al. [52] analyzed a cohort of kidney transplant patients with FD and evaluated the efficacy and safety of ERT (Agalsidase alfa). Among kidney transplant patients receiving ERT, slight increases in serum creatinine as compared to baseline were reported after 2 years. However, proteinuria remained stable. All patients tolerated ERT well with minimal adverse effects. On comparing extra renal effects, the left ventricular mass was greater in untreated patients as opposed to patients on ERT. Similar results were reported in a pilot study from Mignani and colleagues among kidney transplant recipients. Stable renal functions, decreased plasma Gb3 levels, decreased left ventricular mass and improved cardiac contractility were reported among transplanted patients on ERT [64].

One of the challenges with ERT is the development of ERT antibodies, as this complicates treatment among Fabry disease patients [65,66]. In patients after kidney transplant, there appears to be a protective role of immunosuppressive medications on emergence of ERT antibodies. This effect was illustrated by Lenders et al. who evaluated the impact of immunosuppressive medications after kidney transplantation in patients with Fabry nephropathy. Patients who were started on ERT post-kidney transplant did not develop new ERT antibodies in long-term follow up. Concurrently, patients who developed ERT antibodies before transplant had temporary suppression of antibodies post-transplant, indicating the potential protective role of immunosuppression after kidney transplantation [67].

The long-term outcome of kidney transplantation in FD patients was reintroduced after several newer studies reported promising data on graft survival and patient survival. The European Renal Association Registry reported 72% graft survival and 84% patient survival after 3-year follow-up [48]. Similarly, the U.S. Renal Data System has reported a 5-year graft survival and patient survival of 76% and 83%, respectively [54]. The incidence of allograft rejection is identical in patients with vs. without FD, which might indicate that ERT does not affect the rejection rate. In other kidney diseases, rejection is usually caused by under-immunosuppression which could become problematic in immune-mediated diseases where frequent adjustment of immunosuppressive regimen is warranted. Altogether, we showed that kidney transplant in FD patients is not different from other kidney diseases with respect to transplant profile.

Here, we reported the incidence of FD recurrence in kidney transplant patients for the first time (11.1%) with a median follow-up duration ranging from 3 to 11.5 years. Since 1972, the data on recurrent FD post-kidney transplant have been scarcely reported due to small sample sizes. Most studies were retrospective and had quite different biopsy and transplant timings. However, recurrent FD was determined by histological features from the kidney biopsy [54]. Some possible mechanisms leading to recurrent Fabry nephropathy were proposed. New deposition of Gb3 can occur along the endothelial cells within renal tubular cells and podocytes. Moreover, migration of activated macrophage into the graft tissue has also been proposed [26]. More research on the pathogenesis of recurrent Fabry nephropathy in kidney allografts is of great interest as not many studies are currently available.

Our study is the first systematic review to report the allograft outcomes of Fabry nephropathy. However, some limitations may be imposed. First, given the observational design in nature, the reported findings may be subjected to selection bias. Some included studies were published before ERT was established, and it is well known that this treatment had a profound impact on the outcome of the disease. Thus, we performed additionally analysis including only the recent studies (year 2001 or newer when ERT became available) and demonstrated the pooled estimated rates of all-cause graft failure, graft failure before death, and allograft rejection of 28.1%, 11.7%, and 20.2%, respectively. Second, the transplant outcomes of Fabry nephropathy were not directly compared to other diagnoses. Thus, the clinical applicability of our findings should be adopted with caution. Third, although sample sizes were pooled, the total number of subjects remained small. Fourth, the majority of FD patients included in our systematic review were males and data on patients’ comorbidities were limited. Thus, future studies are needed to assess outcomes of kidney transplantation among female patients with FD.

## 5. Conclusions

In summary, the allograft and patient outcomes in patients with FD are comparable to other kidney diseases with a recurrence rate of 11% during a follow-duration of up to 11.5 years.

## Figures and Tables

**Figure 1 diseases-09-00002-f001:**
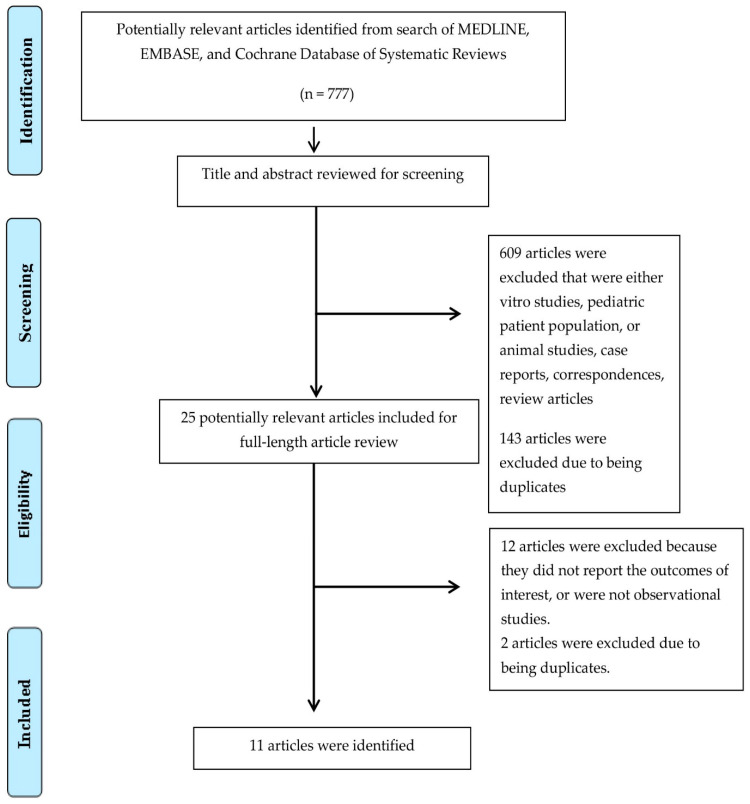
Preferred Reporting Items for Systematic Reviews and Meta-Analysis (PRISMA) flow diagram for study selection.

**Figure 2 diseases-09-00002-f002:**
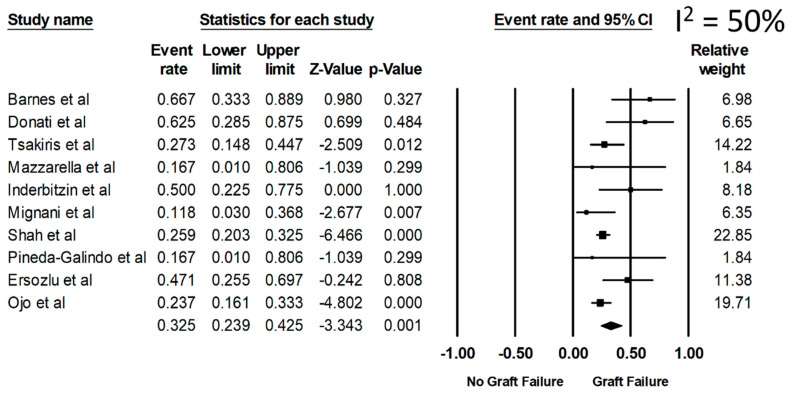
Rate of all-cause graft failure among kidney transplant patients with FD.

**Figure 3 diseases-09-00002-f003:**
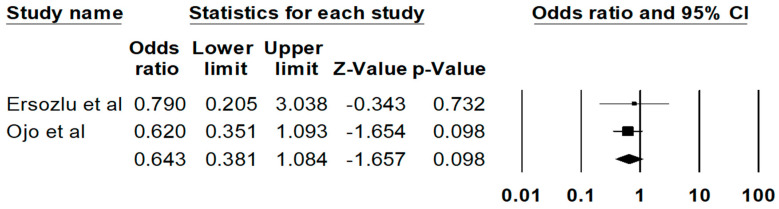
Risk of all-cause graft failure among kidney transplant patients with FD.

**Table 1 diseases-09-00002-t001:** Characteristics of included studies in systematic review [46,47,48,49,50,51,52,53,54,55,56].

Study	Year	N	Male Sex	Mean Age at Diagnosis	Mean Age at Transplant	Enzyme Replacement Therapy	Age at Enzyme Replacement Therapy	Follow-Up Time	Patient Death	Graft Failure Before Death	All-Cause Graft Failure	Graft Rejection	Recurrence of FD in Kidney Allograft
Barnes et al. [46]	1975	9	8/9 (89%)	N/A	41 years	0/0 (0%)	N/A	N/A	6 (67%)	0 (0%)	6 (67%)	N/A	N/A
Donati et al. [47]	1987	8	8/8 (100%)	29.9 years	36.8 years	0/8 (0%)	N/A	3.6 years	0	5 (63%)	5 (63%)	N/A	N/A
Tsakiris et al. [48]	1996	33	73/83 (88%)	N/A	-	0/0 (0%)	N/A	3 years	5 (15%)	N/A	9 (27%)	N/A	N/A
Mazzarella et al. [49]	1997	2	2/2 (100%)	N/A	32 years	0/0 (0%)	N/A	3.8 years	0 (0%)	0 (0%)	0 (0)	0 (0%)	N/A
Inderbitzin et al. [50]	2005	10	10/10 (100%)	26 years	36 years	1/10 (10%)	N/A	10.2 years	4 (40%)	1 (10%)	5 (50%)	1 (10%)	0—clinically 1—in autopsy biopsy
Mignani et al. [51]	2008	17	16/17 (94%)	37.1 years	39.8 years	17/17 (100%)	44.6 years	6 years	0 (0%)	2 (12%)	2 (12%)	N/A	N/A
Cybulla et al. [52]	2009	36	34/36 (94%)	31.1 years	37.6 years	24/36 (67%)	N/A	7.7 years	4 (11%)	3 (8%)	N/A	N/A	N/A
Shah et al. [53]	2009	FD 197 Control 1970	177/197 (90%)	N/A	N/A	N/A	N/A	5 years	FD 37 (19%)	FD 24 (12%)	FD 51 (26%)	FD 41 (21%) Control 528 (27%)	N/A
Ojo et al. [54]	2000	FD 93 Control 186	83/93 (89)	N/A	40 years	N/A	N/A	5 years	FD 16 (17%) Control 34 (18%)	N/A	FD 22 (24%) Control 62 (33%)	N/A	N/A
Pineda-Galindo et al. [55]	2016	2	2/2 (100%)	N/A	N/A	2/2 (100%)	N/A	6 years	0 (0%)	0 (0%)	0 (0%)	N/A	N/A
Ersozlu et al. [56]	2018	FD 17 Control 17	15/17 (88%)	34 years	39.5 years	14/17 (82%)	40.3 years	11.5 years	FD 7 (41%) Control 2 (12%)	FD 2 (12%) Control 9 (53%)	FD 8 (47%) Control 9 (53%)	FD 3 (18%)	FD 2 (12%)—kidney biopsy

Abbreviations: FD, Fabry disease; N/A, Not available; TIA, Transient ischemic attack.

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
