# Peer review of "Outcomes of Kidney Transplantation in Fabry Disease: A Meta-Analysis"

_diseases, 2020, doi:10.3390/diseases9010002_

Round 1
Reviewer 1 Report
Fabry disease (FD) is an important disease for kidney transplantation. medicine. This meta-analysis has a sufficient number of patients and studies, and as conclusions, the fact that there is no difference in the presence or absence of FD, the transplant survival rate, and the patient survival rate is an important fact regarding kidney transplantation.
Author Response
Reviewer 1
Fabry disease (FD) is an important disease for kidney transplantation. medicine. This meta-analysis has a sufficient number of patients and studies, and as conclusions, the fact that there is no difference in the presence or absence of FD, the transplant survival rate, and the patient survival rate is an important fact regarding kidney transplantation.
Response: We thank you for reviewing our manuscript and for your critical evaluation.
We greatly appreciated the editors’ time and comments to improve our manuscript. The manuscript has been improved considerably by the suggested revisions.
Reviewer 2 Report
The authors performed Meta-Analysis on kidney transplantation in Fabry Disease. The sample select method is reasonable, the statistical analysis is logical, the results and MS are present nicely. I don't have any questions for the authors. I suggest accepting the MS.
Author Response
Reviewer 2
The authors performed Meta-Analysis on kidney transplantation in Fabry Disease. The sample select method is reasonable, the statistical analysis is logical, the results and MS are present nicely. I don't have any questions for the authors. I suggest accepting the MS.
Response: We thank you for reviewing our manuscript and for your critical evaluation.
We greatly appreciated the editors’ time and comments to improve our manuscript. The manuscript has been improved considerably by the suggested revisions.
Reviewer 3 Report
This study has several weak points; first the heterogeneity of the studies included; 5 of 11 (for an overall number of 143 patients) were published before enzymatic replacement therapy was established, and it is well known that this treatment had a profound impact on the outcome of the disease. Second the introduction provides an outdated classification of the disease; the stage classification is replaced by the genetic one based on GLA gene mutations; further, the important discovery of Fabry Disease in females is not considered. Data analysis is very limited and we do not have data about sex, replacement therapy (Y/N), co-morbidities, disease duration, interval between clinical onset and therapy. Also the conclusion is weak, the Authors state that clinicians should consider kidney transplantation as an option for renal replacement therapy in patients with end-stage kidney disease secondarily to Fabry Disease; however, clinicians consider this the option whenever the condition you describe occurs. Finally, in the introduction it is reported that current therapies include genetic therapy; however, this is not yet a current therapeutic option for these patients.
Author Response
Reviewer 3
Please see the attachment.

Round 2
Reviewer 3 Report
Thanks for your point to point reply